TOPICAL REVIEW

# Logarithmically scaled, gamma distributed neuronal spiking

Daniel Levenstein[1,2]  and Michael Okun[3] 

[1]*Department of Neurology and Neurosurgery, McGill University, Montreal, QC, Canada*
[2]*Mila, Montréal, QC, Canada*
[3]*Department of Psychology and Neuroscience Institute, University of Sheffield, Sheffield, UK*

Handling Editors: Katalin Toth & Samuel Young

The peer review history is available in the Supporting information section of this article (https://doi.org/10.1113/JP282758#support-information-section).

**Abstract**   Naturally log-scaled quantities abound in the nervous system. Distributions of these quantities have non-intuitive properties, which have implications for data analysis and the understanding of neural circuits. Here, we review the log-scaled statistics of neuronal spiking and the relevant analytical probability distributions. Recent work using log-scaling revealed that interspike intervals of forebrain neurons segregate into discrete modes reflecting spiking at different timescales and are each well-approximated by a gamma distribution. Each neuron spends most of the time in an irregular spiking 'ground state' with the longest intervals, which determines the mean firing rate of the neuron. Across the entire neuronal population, firing rates are log-scaled and well approximated by the gamma distribution, with a small number of highly active neurons and an overabundance of low rate neurons (the 'dark matter'). These results are intricately linked to a heterogeneous balanced operating regime, which confers upon neuronal circuits multiple computational advantages and has evolutionarily ancient origins.

(Received 9 May 2022; accepted after revision 28 July 2022; first published online 10 September 2022)
**Corresponding author** M. Okun: Department of Psychology and Neuroscience Institute, University of Sheffield, Sheffield S10 2TN, UK.    Email: m.okun@sheffield.ac.uk

## Introduction

Somatic spiking is the currency of neuronal communication. Athough spikes used to be recorded from one cell at a time, continuously improving methods now allow simultaneous recording from hundreds to thousands of neurons over multiple hours (Fig. 1*A*). It has been repeatedly pointed out that essential quantities in these data are logarithmically scaled. Key examples

**Daniel Levenstein** completed his PhD at New York University under the guidance of György Buzsáki and John Rinzel. He is currently a postdoctoral researcher at McGill University, working with Adrien Peyrache and Blake Richards. His work focuses on the use of offline, or 'spontaneous', activity to support learning in heterogeneous biological and artificial neural networks. Towards that end, he uses biologically inspired neural network models, as well as neural data analysis, and works closely with experimental collaborators. **Michael Okun** is a Lecturer at University of Sheffield. After obtaining a PhD in theoretical computer science he switched to systems and computational neuroscience research. Michael started his laboratory in 2016. He is interested in cortical circuit dynamics, particularly in the spontaneous activity regime and over long timescales. His lab employs electrophysiological and computational methods to explore the activity of individual neurons and neuronal populations in normal conditions and under the influence of psychoactive compounds.

D. Levenstein and M. Okun contributed equally to this work.

The Journal of Physiology

include firing rates across neurons that have been described as log-normal (Fig. 1*B*) (Buzsaki & Mizuseki, 2014; Hromadka et al., 2008), interspike intervals (ISIs) that span five orders of magnitude (Fig. 1*C*) (Chung et al., 1970), the spectrum of firing rate and field potential fluctuations that follow power law statistics (Buzsaki & Draguhn, 2004; He, 2014; Okun et al., 2019; Teich et al., 1997) and others (Loewenstein et al., 2011; Song et al., 2005). However, this widely observed fact is sometimes underappreciated and has important non-intuitive consequences for both data analysis and interpretation.

Here, we review the log-scaled statistics of spiking in single neurons and neuronal populations. We outline the importance of log-scaling and the empirically observed statistics of ISI and firing rate distributions. We then review the utility of the gamma distribution in describing these empirical data. Although the distribution of ISIs within single neurons and of mean firing rates across neurons are seemingly two disparate phenomena, we argue that, in both cases, the log-scaling is a consequence of a fundamental feature of the nervous system: the need to balance the opposing effects of separate depolarising (excitatory) and hyperpolarising (inhibitory) synaptic currents from distinct groups of presynaptic cells.

## Log-transform

Assumptions of normality abound in data analysis and statistical methods. The normal (Gaussian) distribution has an instantly recognisable bell-shaped probability density function (PDF), with intuitive properties that are often taken for granted. For example, the mean, median and mode of a normal distribution are equal, and most of the data fall close to this value. Thus, when visually examining the PDF, the values of the mean or, for example, the 25th or 90th percentiles are apparent.

Normal distributions are ubiquitous in experimental data as a consequence of the central limit theorem, which roughly states that when the effects of many random, independent processes are combined additively, the result is normally distributed.

However, the distributions of many important quantities are not even approximately bell-shaped. PDFs of such distributions are often less visually informative than their normal counterparts, and they can have counterintuitive properties when approached from a 'bell-shaped distribution' mindset. For example, consider the PDFs in Fig. 2*A*. By visual inspection, it is difficult to answer even basic questions, such as the approximate values of the mean, median and SD of each distribution (mode is not even defined because the PDFs are unbounded for $x \rightarrow 0$) or how the distributions compare. One might assume that one reason for such difficulty is the heavy right tail, which is poorly visualised on a linear scale. Although log-scaling the *x*-axis might help, this is not guaranteed (Fig. 2*B*). On the other hand, if the PDFs of the logarithms of the original values are considered, the picture becomes clear (Fig. 2*C*). In our example, the log-transform makes the distributions bell-shaped (albeit left-skewed) and reveals intuitive differences between the two PDFs. This simple illustration hopefully goes some way towards convincing the reader of the importance of appropriately transforming the quantitative variables under consideration.

For an intuitive understanding of the log-transform, consider that it is equivalent to a histogram in which the size of the bins scales exponentially with *x*. (An illustration using datapoints drawn from the example PDFs and its MATLAB code is provided in the statistical summary document, included as Supporting information). For example, if *x* is duration, the size of bins for datapoints in the millisecond range has a

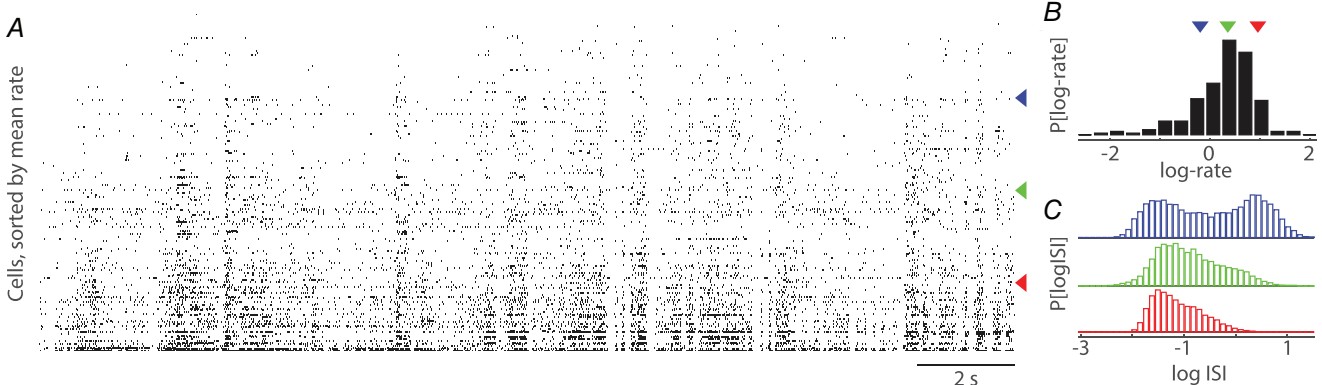

**Figure 1. Spiking activity in a neuronal population**

*A*, activity of 201 neurons in primary visual cortex of a mouse, during 20 s of spontaneous activity. Neurons were ordered by their mean firing rate. *B*, distribution of the log-transformed mean firing rates of the neurons in (*A*). *C*, distribution of log-transformed ISIs (we use base 10 logarithms, unless explicitly stating otherwise) of the three indicated example neurons in (*A*). Based on publicly available multi-hour recording data of Siegle, Jia et al. (2021).

millisecond order of magnitude, whereas, for datapoints in the range of seconds, the bins are also on the scale of a second. Accoridngly, log-transformed data comprise a scale-agnostic representation; for a similar reason, log values are dimensionless (Matta et al., 2011).

## Single neuron spiking

The log-transform is required for basic quantitative spike train analysis, such as the analysis of ISIs. ISIs of a typical neuron *in vivo* can easily range over five orders of magnitude, from milliseconds to minutes. For a distribution spanning many orders of magnitude, the log-transform is necessary, if only to visualise the full range of the data (Fig. 1*C*). On a more conceptual level, as we discuss next, the fact that ISIs are log-scaled reflects two fundamental features of neuronal spiking *in vivo*: irregularity resulting from fluctuating input and the super-position of spiking patterns at multiple timescales.

**Irregular spiking produces log-scaled ISIs.** Although isolated neurons respond to constant current injection with regular spiking (i.e. all ISIs are approximately similar), neurons *in vivo* spike in a highly irregular manner (Burns & Webb, 1976; Compte et al., 2003; Softky & Koch, 1993). The relationship between irregular spiking and log-scaling is already seen in the simplest analytical model of irregular spiking: the homogeneous Poisson process. The inter-event intervals of a Poisson process with rate $r$ are exponentially-distributed, with a mean of $1/r$ and a heavy right tail of long intervals (Fig. 3*A*). The SD of the exponential distribution is also $1/r$, producing intervals that, regardless of rate, have a coefficient of variation of one. Exponential distributions with distinct rates differ in their shape on the linear scale, whereas, upon log transformation, a change in rate corresponds to a translation of the distribution (we omit the formal proof, which is mathematically straightforward) (Fig. 3*A*). Thus, intervals from a Poisson process are naturally log-scaled and such scaling facilitates comparison between processes with different rates.

The Poisson process, however, is only a crude approximation of empirical spike trains. Even under constant input conditions, the probability of a neuron spiking depends on its recent spike history. Such history effects are minimal after long spike-free intervals (the cell has effectively 'forgotten' its last spike time), whereas they are especially prominent in the ∼20 ms following a spike. The relationship between neuronal inputs, refractory effects and the properties of their spiking output is well-captured by the canonical integrate and fire models (Burkitt, 2006). Although these models are highly idealised, they capture the key biophysics of spikes as all-or-none events followed by a quasi-resetting of neuronal state (Hodgkin & Huxley, 1952). For constant inputs, integrate and fire models show a sharp transition from silence to regular spiking at threshold levels of input (Fig. 3*B*), which is comparable to the response properties of many cells *in vitro*. However, when the input includes fluctuations around a constant subthreshold mean, integrate and fire models can produce irregular spiking at *in vivo*-like rates for a wide range of input magnitudes (Fig. 3*B*) (Feng & Brown, 1999; Gerstein & Mandelbrot, 1964; Holt et al., 1996). Spiking in this 'fluctuation-driven' regime is primarily caused by occasional threshold crossings and occurs at a rate determined by the mean and variance of the resulting voltage, relative to the spike threshold.

Fluctuation-driven spiking in integrate and fire models results in ISIs that are well-approximated by a gamma distribution (Fig. 3*C*) (Miura et al., 2007; Ostojic, 2011). Similar to Poisson ISIs, the right tail of gamma-distributed ISIs is exponential, reflecting the steady-state rate set by the input statistics and loss of dependence on spike history at longer timescales. By contrast, the left tail can be supra- or sub-Poisson, reflecting either an increased or decreased probability to spike after the previous spike. Approximately gamma-distributed ISIs are a very general feature of integrate and fire-like systems, and extend to more realistic neuron models, input structure and membrane voltage statistics, including conductance changes caused by presynaptic spikes and input structure other than white noise (Ostojic, 2011).

**Figure 2. Example of the log-transformation**
*A* and *B*, two gamma PDFs plotted on linear and log-scale. *C*, PDFs of the two distributions upon log-transformation.

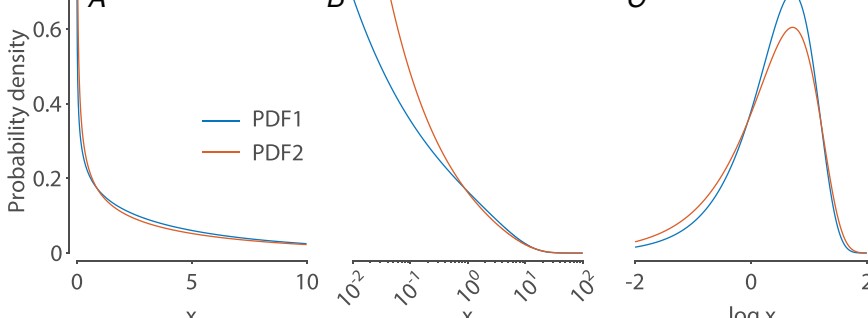

**Table 1. Definitions of log-normal and log-gamma distributions**

| | | |
|---|---|---|
| Log-normal | $X \sim$ Normal$(\mu,\sigma) \Rightarrow \exp(X) \sim$ Log-normal$(\mu,\sigma)$ | $Y \sim$ Log-normal$(\mu,\sigma) \Rightarrow \ln(Y) \sim$ Normal$(\mu,\sigma)$ |
| Log-gamma | $X \sim$ Gamma$(\alpha,\theta) \Rightarrow \ln(X) \sim$ Log-gamma$(\alpha,\theta)$ | $Y \sim$ Log-gamma$(\alpha,\theta) \Rightarrow \exp(Y) \sim$ Gamma$(\alpha,\theta)$ |

Similar to the normal distribution, the gamma distribution has two parameters ($\alpha$ and $\theta$) (Table 1), which can be directly related to the rate of spiking and its irregularity, via the coefficient of variation of the ISIs (rate $= 1/\alpha\theta$, CV $= 1/\sqrt{\alpha}$). On a log scale, gamma-distributed ISIs have many of the intuitive properties that normal distributions have on linear scale: varying rate corresponds to a translation of the distribution, whereas the shape parameter scales the width of distribution. Thus, the statistics of single-cell spiking under stationary input conditions is especially well visualized and compared across conditions on log-scale, in which a gamma distribution of ISIs turns into a log-gamma distribution of log-intervals (Table 1). The distribution of log-scaled ISIs provides an intuitive picture of the relationship between neuronal input and spike output (or transfer function) of the integrate and fire neuron on a log-scale, in which the subthreshold

regime has a linear response at low input magnitudes that saturates at suprathreshold spiking with a narrower ISI distribution (Fig. 3D). The shape of the log-ISI distribution of the Poisson neuron is constant, whereas the shape of the log-ISI distribution from the integrate and fire neuron varies with the mean and variance of its input. With higher mean input, the distribution narrows (Fig. 3C).

**Single neuron spiking as a mixture of gamma PDFs.** Mirroring the integrate and fire models, the ISI statistics of biological neurons under fixed conditions have been well fit by a gamma distribution. *In vitro*, cortical neurons receiving fluctuating current input have ISIs for which the distribution is closely matched by a gamma PDF (Miura et al., 2007). This is also the case *in vivo*, over short windows of spontaneous activity

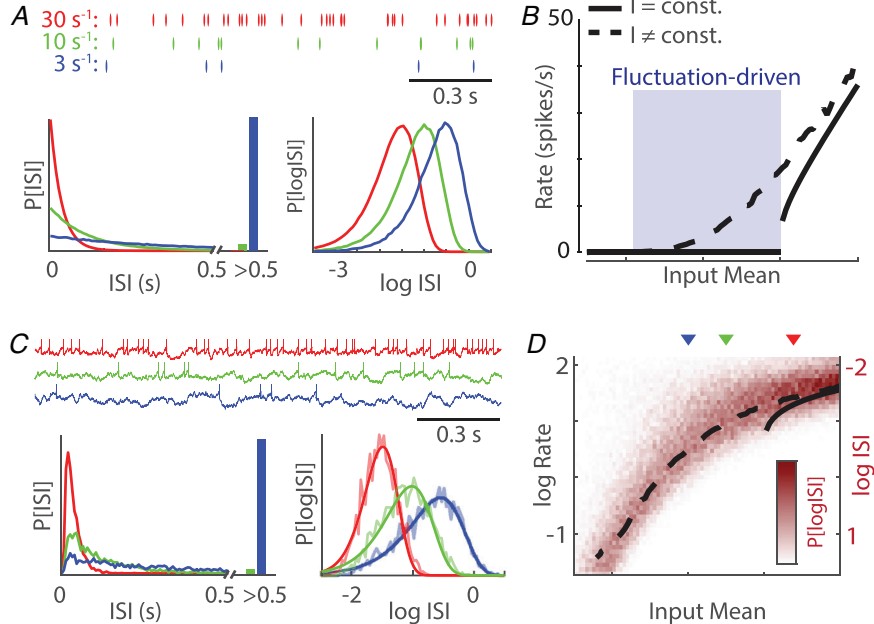

**Figure 3. Log-scaled ISIs of integrate and fire neurons**
*A*, ISI distributions from homogeneous Poisson processes with rate = 30, 10 and 3 spikes s$^{-1}$. On a linear scale, the ISI distributions change shape whereas on a logarithmic scale, different rates correspond to a translation of the distribution. *B*, input/output relationship (transfer function) of a simulated linear integrate and fire neuron with constant and fluctuating inputs. Fluctuation-driven subthreshold spiking regime region is shaded. *C*, integrate and fire model with fluctuating input produces log-scaled spiking statistics. ISI distributions from neurons with different levels of input are readily compared upon log-transformation, but not on a linear scale, and are well-fit by a log-gamma distribution (solid lines, bottom right). *D*, input/output transfer function of an integrate and fire neuron on a log-scale. Heatmap shows the log-ISI distribution as a function of input, with reverse orientation to match rate. Triangles indicate the input values for the integrate and fire models in (*C*).

## Normal, gamma, log-normal and log-gamma distributions

Both gamma and log-normal distributions are naturally log-scaled because their PDFs become bell-shaped upon log-transformation. Therefore, these analytical distributions allow us to translate many of our intuitions from a normally-distributed mindset to analysis of log-scaled spiking data.

Normal and gamma distributions are specified by two parameters. The normal PDF is always bell-shaped and symmetric, with the centre of the bell and its width specified by $\mu$ (mean) and $\sigma > 0$ (standard deviation). Gamma PDF is specified by the shape ($\alpha > 0$) and scale ($\theta > 0$) parameters. The shape of gamma PDF is versatile: for low values of $\alpha$, it looks as illustrated in Fig. 2*A*; for higher values of $\alpha$, it progressively becomes bell-shaped albeit with right (positive) skew. The exponential distribution is a special case of gamma PDF with $\alpha = 1$.

Log-normal and log-gamma distributions involve the normal and gamma distributions and the log-transform. The naming, however, is confusing because the two cases use different nomenclatures. Specifically, log-gamma is the distribution one gets after passing a gamma-distributed random variable through the log-transform. On the other hand, distribution is said to be log-normal if, upon being log-transformed, it has a normal distribution (Table 1).

(Mochizuki et al., 2016) or when the data consist of rate-matched trials of a behavioural task (Maimon & Assad, 2009).

However, a gamma distribution does not provide a good fit to the ISI statistics of most neurons when recorded over long durations and in changing contexts (Fig. 1*C*). This is not unexpected, given the prominence of temporal variation in neuronal firing rate, which is one of the most pervasive and longstanding observations of *in vivo* neurophysiology. For example, accounting for ISIs over long durations in the early visual system requires a non-stationary gamma PDF with a fluctuating, time-dependent rate (Miura et al., 2007; Teich et al., 1997). Such fluctuations in rate also explain the supra-Poisson variability of spiking observed in the cortex (Churchland et al., 2010; Goris et al., 2014) and can be captured by generalized linear models (Gerstner et al., 2014) that combine external sources modulating the probability (or rate) of spiking with spike history effects. Rate-modulating factors can include information from a relevant sensory modality (Pillow et al., 2008; Truccolo et al., 2005) or internal network factors, such as activity of the rest of the local population (Goris et al., 2014; Harris et al., 2003; Lin et al.,

2015; Okun et al., 2015) and oscillations (Hardcastle et al., 2017; McClain et al., 2019).

The regularity of spiking is influenced by multiple factors, the most obvious of which is the discharge rate (Ponce-Alvarez et al., 2010). For example, ISIs from higher rate trials in parietal neurons were on average fit by a gamma PDF with a larger $\alpha$ parameter, indicating more regular spiking (Maimon & Assad, 2009). In other cases, such as thalamic head direction cells, spiking variability was found to be modulated by the animal's head direction in a manner that was not correlated with firing rate (Liu & Lengyel, 2021). Furthermore, neurons often discharge with specific regularity when spiking at different timescales. For example, hippocampal pyramidal cells with highly diverse mean firing rates have notoriously fixed burst ISIs of ∼5 ms (Harris et al., 2001; Ranck, 1973). Recent work analysing recordings from multiple rodent forebrain regions has found that ISI distributions over long duration recordings in freely-behaving animals are well-captured by a mixture model consisting of a small number of gamma PDFs (Fig. 4) (Levenstein et al., 2021). The specific mixture forms a fingerprint of the activity of each neuron, in which each gamma component captures the contribution of a single spiking 'mode' with intervals at a particular timescale and level of variability (Fig. 4*A*).

The mixture of gammas fit reveals that each neuron spends most of the time in a single low rate mode of irregular spiking: the ground state (GS). The increased firing rate during responses to place or head direction in CA1 and thalamic neurons could be attributed to the increased occupancy of specific, discrete activated state (AS) modes with more regular and higher rate spiking, rather than continuously varying rate (Fig. 4*B*). Similar observations have been made in the dimming fibres of the frog's optic nerve in response to specific stimulus features (Chung et al., 1970), as well as in the auditory cortex where ISI distributions show stimulus-evoked changes that are not apparent in firing rate, when AS modes overlap (Insanally et al., 2019). AS modes were seen throughout spontaneous activity (albeit with lower occupancy) and were composed of spike intervals at characteristic timescales similar across neurons (Fig. 4*C*), ranging from very regular (e.g. in bursts with ISIs <10 ms and theta-related spiking with ISIs of ∼100 ms) to more irregular (but still sub-Poisson, e.g. at gamma-oscillation timescales of 30−100 ms). By contrast, the rate of spiking in the GS mode was heterogeneous between neurons (Fig. 4*C*). The GS rate, rather than the propensity of cells to enter activated states, was the main determinant of a cell's mean rate (Fig. 4*D*).

## Population spiking

The previous section discussed the structure of spike trains of single neurons and the origin of the mean firing

rate in the ground state mode of their ISI distribution. In this section, we move to review how this mean rate is distributed in space, that is, across neurons of a brain area of interest. We will see that log-scaling and gamma PDF feature prominently in answering this question as well.

### 'Dark matter' in the cortex and the gamma-distribution of firing rates.

Until the last $10-15$ years, the most widespread approach for recording spiking activity *in vivo* (and in the cortex in particular) utilised single metal microelectrodes. With this method, an experimenter would slowly advance an electrode through the cortex searching for spiking activity. By selecting for neurons exhibiting vigorous spiking, this process produced data biased towards fast-firing neurons. However, the need to search for such neurons at all, given the hundreds of neurons within the $\sim50$ $\mu$m detection radius of the electrode tip (Neto et al., 2016), indicates that fast-firing neurons are not very common. These considerations led to an appreciation that the cortex is significantly more silent than suggested by microelectrode data (Humphries, 2021). It was even suggested that over 90% of all cortical neurons are silent, hence presenting an unobservable 'dark matter' problem in neuroscience (Shoham et al., 2006).

In a seminal study, Hromadka et al. (2008) performed a series of juxtacellular recordings in primary auditory cortex of awake rats with glass recording pipettes, confirming that the majority of neurons had low firing rates, and crucially that the distribution of log-scaled firing rates was approximately Gaussian (i.e. the distribution of firing rates was approximately log-normal) (Fig. 1*B*). This was confirmed by additional juxtacellular and whole-cell recording datasets from multiple cortical areas, as reviewed by Barth & Poulet (2012). Juxtacellular

recordings simultaneously removed the selection bias for fast-firing neurons and failed to find a prevalence of silent neurons. For example, O'Connor et al. (2010) reported that only $\sim13\%$ of the neurons in barrel cortex (14/106) were categorised as silent (firing rate $<0.01$ spikes $s^{-1}$) (Fig. 5*A*). Although these recordings are as close as we now have to 'ground truth' of neuronal spike rates, they are both technically challenging and highly limited in their yield, especially in freely behaving animals.

Over the last decade, microelectrodes have been almost completely superseded by multi-electrode and silicon probes. These devices are inserted 'blindly' into the target brain region, and thus represent a major improvement in terms of selection bias in addition to the higher yield of simultaneously recorded neurons. Firing rates of neuronal populations recorded by such devices were also found to have log-scaled distributions (Buzsaki & Mizuseki, 2014; Mizuseki & Buzsaki, 2013). Directly comparing the distributions of firing rates between silicon probe and juxtacellular data, Fig. 5*B* shows the so-called Lorenz curves for a large dataset of Neuropixels recordings (Siegle, Jia et al., 2021) and for the aforementioned juxtacellular dataset. The Lorenz curves show how the total spike 'budget' is distributed across a neuronal population and visualises the degree of neuronal 'inequality' as the distance of the curve from the diagonal. Although our comparison is not fully like-to-like (the Neuropixels recordings are from visual rather than somatosensory cortex, and the behaviours of mice are distinct), the Lorenz curves demonstrate that both datasets have a highly non-uniform distribution of firing rates, with a majority of spikes produced by a small subset of neurons. However, similar to the microelectrode data, the silicon probe data have fewer slow-firing neurons compared to juxtacellular recordings. Specifically, the slow-firing half of neurons

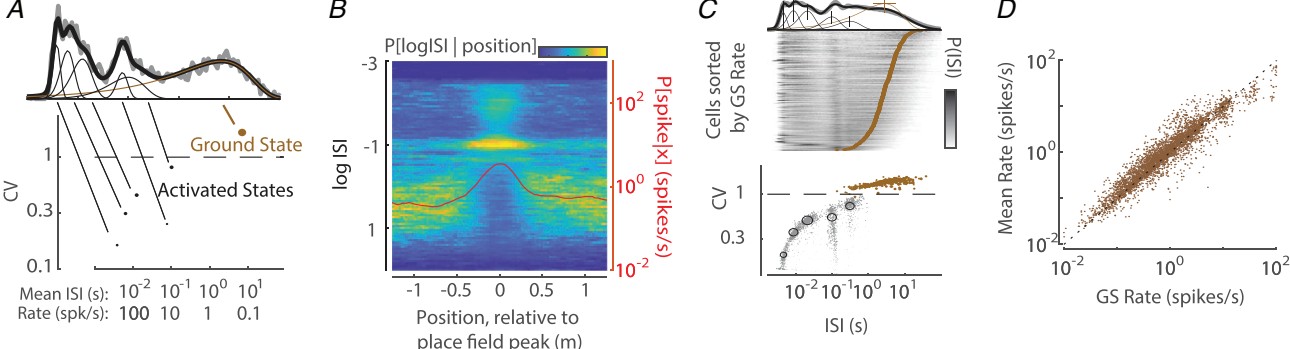

**Figure 4. ISI distribution as a mixture of gamma distributions**
*A*, decomposition of an ISI distribution of an example CA1 neuron. The empirical ISI distribution is decomposed into a mixture of six gamma distributions (GS mode and five AS modes), each with specific shape, scale and weight (proportion of ISIs, indicated by the size of the corresponding circle). *B*, ISI distribution conditioned on position relative to place field peak, averaged over place cells recorded from CA1 area. *C*, ISI distribution of 562 CA1 neurons and its gamma decomposition (neurons are sorted by their GS rate). *D*, rate of the GS mode is highly correlated with mean rate. Points reflect neurons across multiple brain areas, including thalamus, cortex, amygdala and hippocampus. Adapted from Levenstein et al. (2021).

accounts for 8.7% of all the spikes in Neuropixels data, but only for 4.1% in the juxtacellular data (Fig. 5*B*). The bias towards fast-spiking neurons in silicon probe data is significantly less severe than for microelectrodes, but stems from the same factors: completely silent neurons are not detected, and neurons that only fire a small number of spikes are challenging to isolate with spike-sorting software. The existence of a bias towards fast-spiking neurons is supported by an anatomical estimate of the number of neurons within the detection radius of the probe, which suggests that ∼2-fold more neurons should have been recorded (Siegle, Ledochowitsch et al., 2021). These extra neurons, however, are not completely missing from silicon probe data. In addition to spikes of the well isolated neurons, such recordings contain a hash of spikes for which waveforms cannot be isolated into clearly distinct clusters (Rossant et al., 2016; Trautmann et al., 2019). This so-called multi-unit activity typically has ∼20–50% of the amount of single neuron spikes. Thus, the fact that anatomically there are twice as many neurons as are isolated electrophysiologically can be easily accounted for by multi-unit activity spikes, if it is presumed that the non-isolated neurons have low firing rates (e.g. on par with the slow-spiking half of the well-isolated neurons).

How can we describe the population-wide distribution of neuronal mean firing rates? A decade ago, the fit quality of log-normal and gamma PDFs was compared in eight datasets of mainly tetrode recordings, finding an almost equal split (5/8 *vs*. 3/8) between the two (Wohrer et al., 2013). Here, we revisited this question using juxta-cellular and Neuropixels recordings. For the juxtacellular recordings, we examined how gamma and log-normal distributions are able to fit the Lorenz curve plot for non-silent neurons in Fig. 5*B*. As shown in Fig. 6*A* and *B*, the gamma distribution provides a close fit to the empirical curve, whereas the log-normal distribution does

not. In agreement, the gamma PDF provides a better fit than the log-normal one for the Neuropixels recordings of the multiple brain areas examined, as detailed in Dearnley et al. (2021). It is worth noting that we do not claim that gamma is 'the right' analytical distribution to use, but rather that it is relatively accurate and convenient to fit. It is possible that other analytical PDFs could provide an even better fit, particularly if specified by >2 parameters (e.g. the generalised gamma distribution). A concise description afforded by a closely fitting analytical PDF allows easy quantitative comparisons of the distribution between brain areas or states, and could provide important insights into the operational regime of the neuro-nal network and the computation and communication strategies it implements.

To understand why a gamma PDF performs better than a log-normal PDF, log-transforming the data proves to be useful again. We find that the log-gamma PDF accurately captures the left-skewness of the empirical log-rate data, whereas a Gaussian cannot (Fig. 6*C* and *D*). The left skew indicates a prevalence of slow-firing neurons beyond what is captured by a log-normal distribution. The better fit to the empirical data provided by the gamma distribution also underscores the importance of log-transforming the firing rates. For the log-normal distribution, simple logarithmic rescaling of the *x*-axis is sufficient to attain a bell-shaped PDF, but, as mentioned above, this is not the case for the gamma distribution (Fig. 2*B*; see also Supporting Information). Therefore, the transformation to log-rates cannot be avoided if the aim is to examine the population firing rates on their natural scale.

These results suggest that the 'dark matter' problem is a non-problem. The majority of the neurons simply have low firing rates as a result of the overall log-scaled distribution with a heavy left tail, and the previous inability to identify their spikes was a consequence of

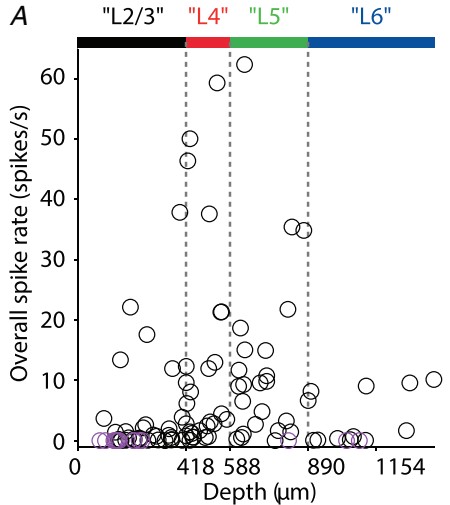
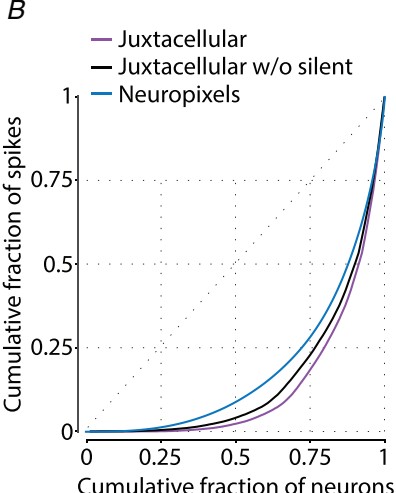

**Figure 5. Distribution of firing rates across cortical neuronal population**
*A*, summary of the juxtacellular dataset showing the cortical depth and firing rate of each cell. Silent neurons (firing rate <0.01 spikes s$^{-1}$) are shown in purple. *B*, cumulative fraction of neurons (ordered from lowest to highest firing rate) *vs*. cumulative fraction of spikes, for three datasets; purple, black: juxtacellular dataset of (O'Connor et al., 2010) with all the neurons (purple) or with non-silent neurons only (black); blue: Neuropixels Allen Institute dataset (from 22 recordings in the primary visual cortex, similar to the one illustrated in Fig. 1*A*). Adapted from O'Connor et al. (2010).

this. The results presented by O'Connor et al. (2010), which, based on the above arguments, appear to provide a good estimate of cortical firing rates, together with the fit of the non-silent subset by gamma PDF ($\alpha \approx 0.38$ and $\theta \approx 22.3$, $\theta$ was inferred from the median firing rate) (Fig. 6*A*), lead to the following estimate of firing rates in a column of the mouse sensory cortex: ∼15% of the neurons are silent, that is, fire at <0.01 spikes s$^{-1}$, ∼25% fire at 0.01–1 spikes s$^{-1}$ and ∼25% are fast-firing cells with rates >10 spikes s$^{-1}$. Imaging methods hold the promise to

corroborate this estimate in an unbiased manner because each neuron can be seen irrespective of its level of activity. Although presently used calcium indicators do not provide a single spike resolution, the distribution of calcium transients in neuronal populations is also log-scaled (Margolis et al., 2012; Yang et al., 2019; Zarhin et al., 2022). With ongoing improvements of calcium and voltage indicators, it is probable that optical methods will shed definitive light on the 'dark matter' issue over the next decade.

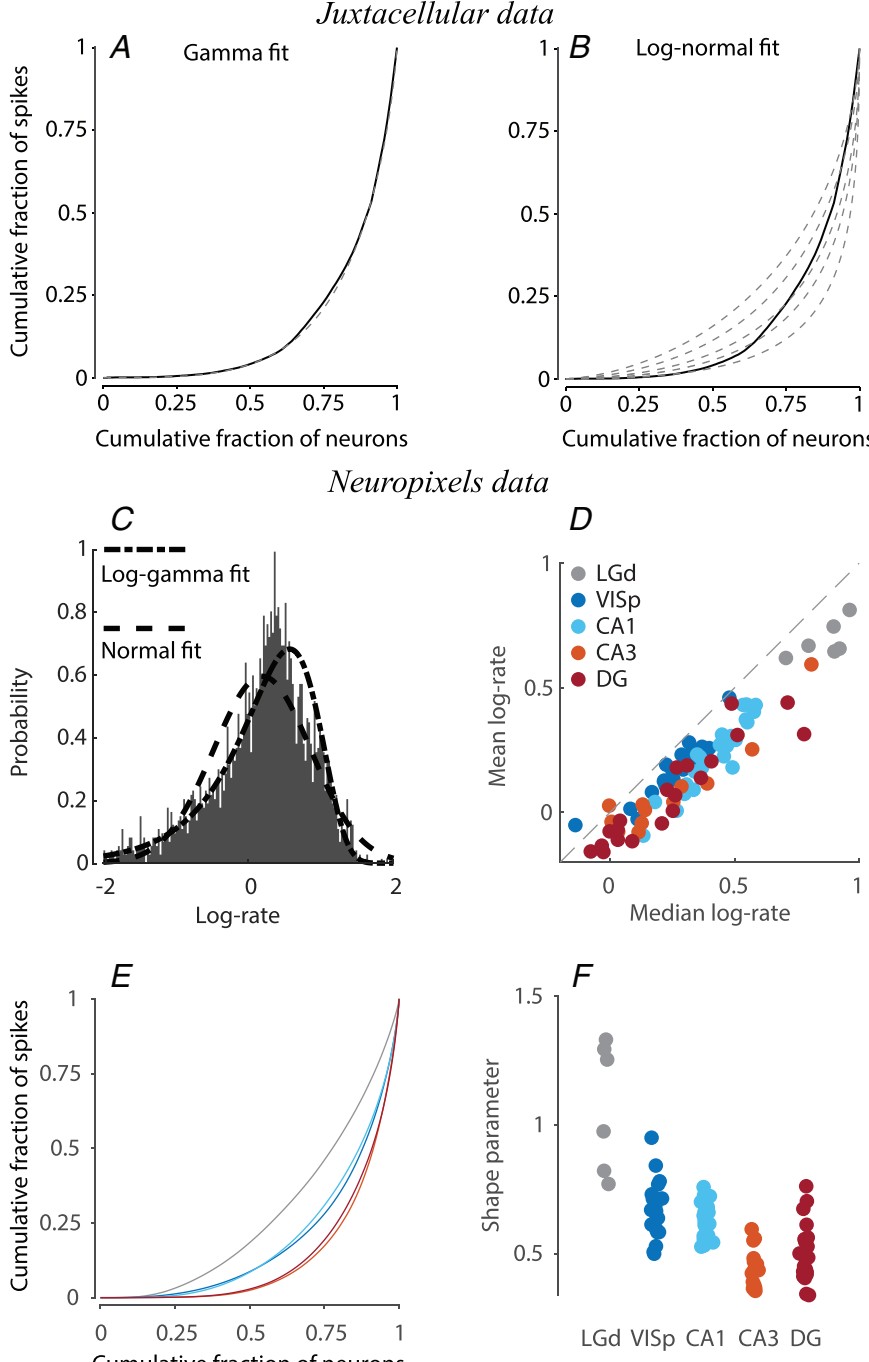

**Figure 6. Empirically observed firing rate distributions and their analytical fitting**

*A*, empirical Lorenz curve (replotted from Fig. 5*B*) is well fit by the gamma distribution (dashed line). *B*, log-normal PDFs do not provide a good fit to this empirical data: dashed lines show the Lorenz curves for several $\sigma$ values within the optimal range, none of them closely fit the empirical Lorenz curve. *C*, log firing rate distribution from Neuropixels recordings in the primary visual cortex is left-skewed and hence better fit by the log-gamma PDF. *D*, across multiple recordings and brain areas, the population log-rate mean is lower than the median, indicating that the log-rate is left-skewed. *E*, different brain areas have distinct Lorenz curves. Colours as in (*D*). *F*, gamma fit to individual recordings shows that the shape of the firing rate distribution is distinct across brain areas (all the differences are statistically significant at $P < 0.001$ using a rank sum test, except VISp *vs*. CA1 and CA3 *vs*. DG). (*C*) to (*F*) are based on the Neuropixels Allen Institute dataset.

**Population rate distribution across distinct brain areas and brain states.** The distribution of mean firing rates differs across brain areas (Mizuseki & Buzsaki, 2013). The difference is not only in the average firing rate, which, for example, is higher in sensory thalamus than in the cortex (Fig. 6*D*), but also in the way the over-all spike 'budget' is spread across neurons. When we compare the Lorenz curves of the different brain areas (Fig. 6*E*), we find that the firing rate distribution is most uniform in the thalamus and most unequal in the dentate gyrus and CA3 hippocampal areas. Conveniently, for gamma-distributed data, the Lorenz curve is controlled by the $\alpha$ shape parameter alone. Therefore, the differences in inequality of firing rate distribution across areas can be easily verified by fitting gamma PDF to individual recordings and comparing the $\alpha$ parameter (Fig. 6*F*).

Recent evidence demonstrates that the population-wide distribution of firing rates is not fixed. Changes in firing rate on timescales of tens of seconds and minutes are partially explained by transitions between different brain states. Often brain state transitions involve changes in the shape of the firing log-rate distribution, which can get translated and squeezed or stretched. For example, the log-rate distribution gets stretched upon transitions from non-REM to REM sleep (Miyawaki et al., 2019; Mizuseki & Buzsaki, 2013; Watson et al., 2016) or from low to high level of arousal in the awake condition (Dearnley et al., 2021). Although the overall distribution of firing rates across the entire population changes across brain state transitions, the rank of individual neurons in the firing rate distribution remains (relatively) conserved, such that fast (slow) firing neurons in one state (e.g. sleep) tend to maintain a high (low) rate in another (e.g. wakefulness) (Buzsaki & Mizuseki, 2014; Dearnley et al., 2021; Hengen et al., 2016; Watson et al., 2016). This is primarily attributed to stable GS rate across brain states, rather than the propensity of cells to enter AS modes (Levenstein et al., 2021). Fitting of an analytical PDF such as gamma can be applied to firing rates of specific neuronal subsets, such as neurons forming a particular cell-type or sharing a response property. Such decomposition of the distribution of firing rates across the entire population into a mixture model (an approach we have already encountered for ISIs) can help elucidate the way that neuronal firing rates change across brain states (Dearnley et al., 2021).

Visualisation and understanding of changes in population firing rates are assisted by the log-transform and by an accurate fit of empirical data. In the example shown in Fig. 2, we see two PDFs with equal means, and the difference between them only became apparent upon log-transformation. This was not merely a hypothetical example using two arbitrary distributions, but comprised the firing rate distributions that we have experimentally observed in prefrontal cortex in two different brain states (Dearnley et al., 2021). This example therefore demonstrates that log-transforming the firing rate data might be crucial for its proper interpretation.

## Log-scaled ISIs and firing rates from balanced input

In the previous sections, we have seen that both ISIs and firing rates are accurately captured as logarithmically scaled, gamma distributed quantities or mixtures thereof. What properties of neurons and neuronal circuits might result in these log-scaled distributions, and what can they tell us about the operating regime of neuronal populations?

Neuronal circuits almost universally follow Dale's principle, according to which depolarizing and hyper-polarizing synaptic currents originate in non-overlapping populations of excitatory (*E*) and inhibitory (*I*) cells. In such circuits, the fluctuation-driven regime necessary to produce irregular spiking at the experimentally observed low rates requires that neurons' *E* and *I* inputs are, on average, 'balanced' in magnitude (Bell et al., 1995; Brunel & Hakim, 1999; Gerstein & Mandelbrot, 1964; Holt et al., 1996; Shadlen & Newsome, 1994, 1998; Stein, 1965; Tiesinga et al., 2000). Highly correlated *E* and *I* inputs are widely observed *in vivo*, during both spontaneous and sensory-evoked conditions (Arroyo et al., 2018; Okun & Lampl, 2009). Balanced inputs emerge naturally in recurrent networks of *E* and *I* neurons with a self-sustaining 'balanced state' in which neurons fire asynchronously and irregularly at low rates (Brunel, 2000; Kumar et al., 2008). Conditions for the balanced state are minimal: *I* synapses need to be sufficiently strong to counteract the positive feedback from self-excitation (Brunel, 2000) and synaptic weights should scale with the number of inputs to a neuron such that fluctuations are sufficiently large to bring neurons across threshold (Kadmon & Sompolinsky, 2015; van Vreeswijk & Sompolinsky, 1996). Experimental evidence suggests that both of these conditions are met, and may even be homeostatically maintained in neuronal circuits on a cell-by-cell basis (Barral & Reyes, 2016; Froemke et al., 2007; Xue et al., 2014). Even non-recurrent inputs can be balanced as a result of the ubiquitous effects of feed-forward inhibition (Bhatia et al., 2019; Buzsaki, 1984).

Log-scaled distributions of ISIs and mean firing rates emerge naturally in networks of balanced neurons. Where ISIs are log-scaled because of the irregular fluctuation-driven spiking, firing rates are naturally log-scaled because fluctuation-driven neurons have a supralinear relationship between mean input and firing rate output (*I/O* transfer function) (Hansel & Vreeswijk, 2002; Miller & Troyer, 2002; Priebe & Ferster, 2008). A supralinear *I/O* transfer function turns a normally distributed variation in membrane potential across

neurons (e.g. via central limit addition of many sources of input) into a distribution of mean firing rates over neurons with a heavy right tail (Roxin et al., 2011). This mechanism was validated experimentally (Petersen & Berg, 2016) and even captures the fact that the log-transformed distribution is left-skewed (Roxin et al., 2011).

The fact that mean firing rate is primarily determined by the GS rate suggests that GS mode is attributable to a stable balanced state of neuronal inputs (Hennequin et al., 2018) and that cell-to-cell variation in GS rate reflects underlying heterogeneity in the subthreshold voltage at which cells balance relative to spike threshold (or 'balance point') (Fig. 7). Such heterogeneity could have multiple sources, such as variation in the influence of local connectivity (or in-degree) (Landau et al., 2016; Okun et al., 2015; Trojanowski et al., 2021), excitability (Sweeney et al., 2015; Trojanowski et al., 2021), or relative strength of inhibitory and excitatory synapses (Vegué & Roxin, 2019; Yassin et al., 2010).

On relaxing the mathematically convenient assumptions of an infinite number of basic integrate and fire neurons with uniform random connectivity receiving a homogeneous input, deviations from constant rate irregular spiking are quickly found that reproduce a variety of AS mode spiking patterns observed *in vivo*. Balanced networks can show a heterogeneous mix of subthreshold and suprathreshold spiking (Bi et al., 2021), which can propagate internally through strong recurrent connections (Omura et al., 2015; Ostojic, 2014) or be evoked by upstream inputs (Vogels & Abbott, 2009). In addition to asynchronous irregular activity, the interaction between excitatory and inhibitory cells can produce coherent (*E/I* gamma) oscillations (Brunel, 2000;

Buzsaki & Wang, 2012) that constrain spiking statistics of individual neurons. Intrinsic neuronal currents also shape spiking at a wide range of timescales, affecting the temporal properties of coherent network activity (Lundstrom et al., 2008; Stark et al., 2013), as well as the timing of responses of single neurons (Larkum et al., 1999; Robinson & Siegelbaum, 2003). Together, these results suggest that, although the irregular spiking GS mode results from balanced input, the repertoire of AS modes results from a variety of cellular and network activity patterns that arise from deviations from the idealised balanced integrate and fire state (Fig. 7). Where GS rate reflects cell steady-state subthreshold membrane potential, the particular temporal properties of different AS modes are determined by network-specific activity regimes and perturbations, which can produce spiking at a range of timescales and ranging from regular (Ranck, 1973) to irregular (Compte et al., 2003). Because the timescales of these activity patterns themselves tend to be distributed over a logarithmic scale (Buzsaki & Draguhn, 2004), this variation in spiking patterns, over time and between cells, can only be compared on a log-scale.

## Computational advantages of log-scaled spiking statistics

What are the benefits of an operational regime in which log-scaled ISIs and firing rates emerge from the need to maintain, on average, low-rate activity in a neuronal network with separate *E* and *I* inputs? One obvious benefit of this regime is energy conservation through low spike rates. Additional computational benefits are discussed next.

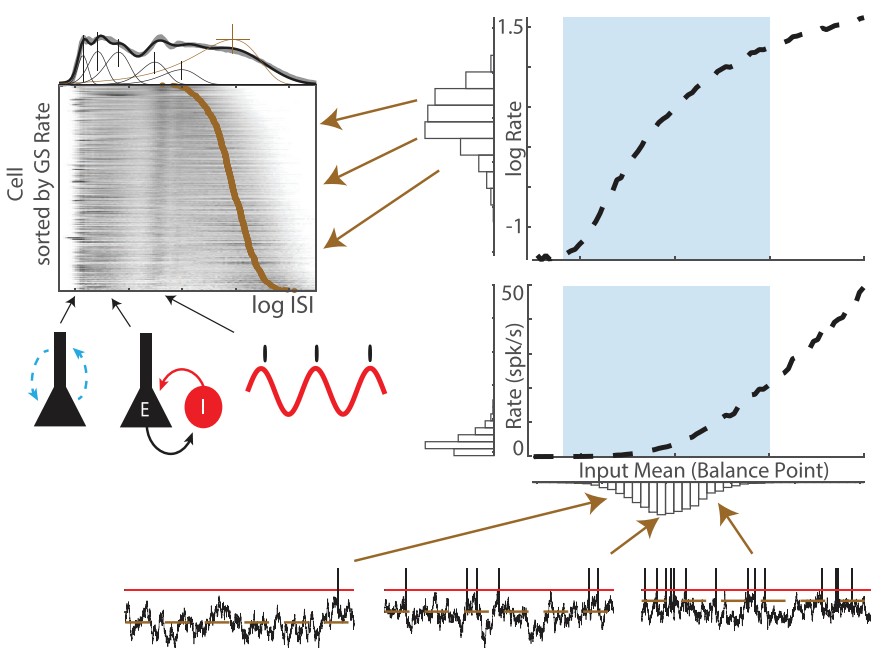

**Figure 7. Log-scaled spiking in *E–I* networks**
The irregular spiking GS mode results from a steady state of fluctuation-driven spiking at heterogeneous rates, which are gamma distributed as a result of normally distributed subthreshold balance points and the supralinear *I/O* transfer function of balanced fluctuation-driven neurons. The repertoire of AS modes results from a variety of cellular and network activity patterns that produce spiking at particular timescales; for example, dendritic burst spiking that produces ~5-ms ISIs, *E–I* 'gamma' oscillations that produce ~10–30-ms ISIs and theta oscillations that produce ~120-ms ISIs. Multiple other mechanisms are not illustrated.

On the single-cell level, it is traditionally assumed that an irregular Poisson spike train has the highest information rate per spike at a given firing rate (Tiesinga et al., 2000) and balanced input may improve the efficiency of neural coding by maintaining the firing irregularity of cortical neurons (Miura et al., 2007). However, although the GS mode is universally irregular, AS spiking modes tend to be more regular, and are heterogeneous across regions (Levenstein et al., 2021). This regional diversity of firing patterns is conserved across mammalian species (Mochizuki et al., 2016) and relates to the functional category of the cortical area; from more regular spiking in motor areas in comparison to sensory areas, and bursty in the prefrontal and hippocampal areas, suggesting that spiking at specific timescales might play a role in distinct neural computations in each functional subdivision (Mochizuki et al., 2016; Shinomoto et al., 2009). Activity at multiple distinct timescales can support functional multiplexing of signal propagation between brain regions (Tingley et al., 2018), and can engage functionally distinct synaptic (Bienenstock et al., 1982) and intracellular (Payeur et al., 2021) processes.

On the population level, networks with log-scaled firing rates are assumed to support a balance of network stability and flexibility. In plastic networks, even a small variability in neuronal parameters can result in a neuronal oligarchy where a small group of interconnected neurons has an exceptionally strong impact on the network dynamics (Kleberg & Triesch, 2018). High rate hubs in these networks can support local signal transmission along sequences of specific subnetworks (Jahnke et al., 2014), and they can sustain a large-amplitude response to transient stimuli, which does not occur in more homogeneous networks (Vegué & Roxin, 2019). Heterogeneous networks with log-scaled firing rates are relatively insensitive to changes in properties of the many low firing rate neurons, which allows them to be plastic without having large effects on the stability of overall network behaviour (Panas et al., 2015). This can support functional segregation by mean firing rate in memory representations (Gava et al., 2021; Grosmark & Buzsaki, 2016) and tuning properties (Lee et al., 2020) that optimize storage (Pereira & Brunel, 2018) and coding capacity (Padmanabhan & Urban, 2010).

**Evolutionary perspective.** Of course, eventually, any appeal to normative properties of biological systems is an appeal to evolution. Although this review primarily focused on the firing rates of neo- and archicortical neurons and neuronal populations, the origins of the phenomena described here appear to be significantly more evolutionarily ancient and widespread.

Several years ago, Berg and colleagues investigated how spikes and membrane potentials are distributed within and across a neuronal population using intracellular and silicon probe recordings (Berg et al., 2007; Lindén & Berg, 2021; Petersen & Berg, 2016). The recordings in these studies were made not in the mammalian cortex but in isolated turtle spinal cord sections, and examined activity that, in the intact animal, would have generated a scratching movement by the hindlimbs. These investigations revealed that the activity of the spinal cord neuronal network exhibits all the key features reviewed in the present review, namely log-scaled firing rates, irregular spiking activity of individual neurons and *E/I* balance.

The spinal cord circuitry is highly evolutionarily conserved. Its comparison across different vertebrate phyla suggests that the neuronal circuits studied by Berg and colleagues are ∼420 million years old (Grillner & El Manira, 2020). This is a time when sharks separated from the line leading to mammals and long before the evolutionary appearance of the mammalian cortex, which is further underscored by the fact that in this network the primary inhibitory neurotransmitter is glycine rather than GABA. This evolutionary perspective, along with the computational considerations outlined in the previous section, indicate that the properties reviewed here are significantly older and more basic than one might infer from the cortex-focused literature that presently dominates the field.

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

## Additional information

### Competing interests

The authors declare that they have no competing interests.

### Author contributions

D.L. and M.O. contributed equally and approved the final version of the manuscript.

### Funding

M.O. was supported by the Academy of Medical Sciences and Wellcome Trust (Springboard award SBF002\1045) and BBSRC (grant BB/P020607/1).

### Acknowledgements

We thank Roman Huszar, Hannes Saal, Rachel Swanson and Brendon Watson for comments on this manuscript. For the purpose of open access, the authors have applied a Creative Commons Attribution (CC BY) licence to any Author Accepted Manuscript version arising.

### Keywords

cortical dark matter, excitation–inhibition balance, firing rate distribution, fluctuation-driven regime, interspike intervals, irregular spiking, log-normal, spike train analysis

## Supporting information

Additional supporting information can be found online in the Supporting Information section at the end of the HTML view of the article. Supporting information files available:

**Statistical Summary Document**
**Peer Review History**

