## [Peer Review History · The Journal of Physiology]

Logarithmically scaled, gamma distributed neuronal spiking

Daniel Levenstein and Michael Okun
DOI: 10.1113/JP282758

Corresponding author(s): Michael Okun (m.okun@sheffield.ac.uk)

The following individual(s) involved in review of this submission have agreed to reveal their identity: Rune Berg (Referee #1); Alex Roxin (Referee #2)

Review Timeline:

Submission Date:	09-May-2022
Editorial Decision:	08-Jun-2022
Revision Received:	23-Jul-2022
Accepted:	28-Jul-2022

Senior Editor: Katalin Toth

Reviewing Editor: Samuel Young

Transaction Report:

Dear Michael,

Re: JP-TR-2022-282758 "Logarithmically scaled, gamma distributed neuronal spiking" by Daniel Levenstein and Michael Okun

Thank you for submitting your Topical Review to The Journal of Physiology. It has been assessed by a Reviewing Editor and by 2 expert referees and I am pleased to tell you that it is considered to be acceptable for publication following satisfactory revision.

The reports are copied at the end of this email. Please address all of the points and incorporate all requested revisions, or explain in your Response to Referees why a change has not been made.

NEW POLICY: In order to improve the transparency of its peer review process The Journal of Physiology publishes online as supporting information the peer review history of all articles accepted for publication. Readers will have access to decision letters, including all Editors' comments and referee reports, for each version of the manuscript and any author responses to peer review comments. Referees can decide whether or not they wish to be named on the peer review history document.

I hope you will find the comments helpful and have no difficulty in revising your manuscript within 4 weeks.

Your revised manuscript should be submitted online using the links in Author Tasks Link Not Available. This link is to the Corresponding Author's own account, if this will cause any problems when submitting the revised version please contact us.

You should upload:

- A Word file of the complete text (including any Tables);
- An Abstract Figure, (with accompanying Legend in the article file)
- Each figure as a separate, high quality, file;
- A full Response to Referees;
- A copy of the manuscript with the changes highlighted.
- Author profile. A short biography (no more than 100 words for one author or 150 words in total for two authors) and a portrait photograph of the two leading authors on the paper. These should be uploaded, clearly labelled, with the manuscript submission. Any standard image format for the photograph is acceptable, but the resolution should be at least 300 dpi and preferably more.

- A 'Cover Art' file for consideration as the Issue's cover image;
- Appropriate Supporting Information (Video, audio or data set https://jp.msubmit.net/cgi-bin/main.plex?form_type=display_requirements#supp).

To create your 'Response to Referees' copy all the reports, including any comments from the Senior and Reviewing Editors into a Word, or similar, file and respond to each point in colour or CAPITALS. Upload this when you submit your revision.

I look forward to receiving your revised submission.

Best wishes

Katalin Toth
Senior Editor
The Journal of Physiology

EDITOR COMMENTS

Reviewing Editor:

The authors have done an excellent job of writing a topical review on scaling of neuronal spiking. Both reviewers agreed that the review was timely and addressed an important topic in the field.. Both reviewers just had extremely minor issues that can be addressed with minor changes to the text. Please rewrite text taking into careful consideration of these points.

REFEREE COMMENTS

Referee #1:

In this manuscript, the authors address the scaling of distributions of neuronal spiking within single neurons as well as across populations. Based on analyses of available experimental data and models they argue that a gamma distribution of firing rates is a better description than lognormal. They discuss "dark matter", irregular/ fluctuation-driven spiking, potential mechanisms of the distribution shape, and other interesting issues. The manuscript is somewhere in between a literature review and novel findings. Given the importance of the topic and the relatively sparse literature on the topic, I see the manuscript as an important contribution. I have a few comments.

Section 2 is important but confusing:

"In our example, the log-transform makes both distributions bell-shaped (albeit left-skewed), revealing that the modes of the two log-distributions are similar but the orange distribution has heavier tails. "

First, there is no orange distribution in figure 2.

What is the difference between 2B and 2C? Some more text in the caption and in the section would be helpful. I recommend writing more text details on the difference and how the measurements were analyzed to give the difference between these two panels. Is the data computer generated?

Regarding:

Section 5: This section is important for explaining the mechanisms behind the skewed distributions. The argument is that a supra-linear I/O function turns normally distributed input (membrane potential) into firing rates that are not normally distributed (due to the nonlinearity). An argument suggested by Roxin et al, as the authors are well aware. The authors primarily justify this assertion by referring to previous modeling papers. But why not include the experimental work of Petersen and Berg 2016 in these references? Petersen and Berg investigated exactly this in the neuronal network dynamics by measuring membrane potential fluctuation and the supra-linearity across the population and the log-scaling of the spiking. They found that the input distribution is Gaussian and on average 3 sigmas from the threshold, both on single-cell level but also across the population.

6.1 "archi-cortical" -> "archicortical".

Referee #2:

In the topical review "Logarithmically scaled, gamma distributed neuronal spiking", the authors discuss findings on the distributions of firing rates and inter-spike intervals (ISI) in, for the most part, cortical neurons. The main thrust of the review is that these distributions are far from Gaussian and are best considered on a logarithmic scale. Overall I believe this is a clear and readable introduction to the topic for the non-expert. This discussion of the differences between Gaussian and log-distributed (log-normal or log-gamma) distributions is well done, as is the discussion of the data. My only concerns would be regarding the details of mechanisms as understood through analysis of single-cell models and networks. I feel some of the details, which I think are of interest, are glossed over.

1 - Relationship between firing rate and CV: I feel Fig.3 could be explained more clearly. In particular, it would be nice to highlight the difference between the Poisson neuron and the LIF as far as the shape of the CV distribution for difference mean rates. As the authors state, the Poisson process always has CV=1, but this is not the case for the LIF and the ISI distribution narrows for increasing rates, a very general property of IF models and spiking networks in general. This is actually an open challenge in neuroscience since the CV for neurons in-vivo remains high even in the AS, e.g. Compte, Constantinidis et al. J. Neurophysiology 2003.

2 - Dale's principle and the fluctuation-driven regime: In section 5 the authors state that log-scaled ISIs and firing rates are somehow a consequence of Dale's Law. I have to say I don't understand this point. Dale's law states that each neuron is either E or I and cannot have an excitatory effect on some targets and an inhibitory effect on others. From the point of view of the post-synaptic neuron, the requirement for being in the fluctuation-driven regime is that E and I inputs should cancel more or less in the mean. This mechanism is agnostic as to where these inputs are coming from, and hence does not

depend on Dale's law. At the network level, to find a self-consistent network state in a recurrent circuit in the fluctuation-driven regime may in some way depend on Dale's law, but it's not clear to me how. Once the network is in the fluctuation-driven regime then log-scaled rates and ISIs follow, yes.

3 - Left-skewed log-rate distributions: The authors note that actual firing rate distributions are not exactly log-normal, but rather they have fatter left tails on a log-scale. They show that this is well fit by a log-gamma. It may also be of interest to show that the firing rate distributions calculated analytically from the theory for LIF neurons as well as the results from simulations of spiking neurons also reproduce this left-skewness and, in fact, can fit in-vivo data quite well, e.g. Roxin et al. J. Neurosci. 2011 Fig7.

REQUIRED ITEMS:

-Please include an Abstract Figure. The Abstract Figure is a piece of artwork designed to give readers an immediate understanding of the Review Article and should summarise the main conclusions. If possible, the image should be easily 'readable' from left to right or top to bottom. It should show the physiological relevance of the Review so readers can assess the importance and content of the article. Abstract Figures should not merely recapitulate other figures in the Review. Please try to keep the diagram as simple as possible and without superfluous information that may distract from the main conclusion of the Review. Abstract Figures must be provided by authors no later than the revised manuscript stage and should be uploaded as a separate file during online submission labelled as File Type 'Abstract Figure'. Please ensure that you include the figure legend in the main article file. All Abstract Figures will be sent to a professional illustrator for redrawing and you may be asked to approve the redrawn figure before your paper is accepted.

-Your MS must include a complete "Additional information section" with the following 4 headings and content:

Competing Interests: A statement regarding competing interests. If there are no competing interests, a statement to this effect must be included. All authors should disclose any conflict of interest in accordance with journal policy.

Author contributions: Each author should take responsibility for a particular section of the study and have contributed to writing the paper. Acquisition of funding, administrative support or the collection of data alone does not justify authorship; these contributions to the study should be listed in the Acknowledgements. Additional information such as 'X and Y have contributed equally to this work' may be added as a footnote on the title page.

It must be stated that all authors approved the final version of the manuscript and that all persons designated as authors qualify for authorship, and all those who qualify for authorship are listed.

Funding: Authors must indicate all sources of funding, including grant numbers. If authors have not received funding, this must be stated.

It is the responsibility of authors funded by RCUK to adhere to their policy regarding funding sources and underlying research material. The policy requires funding information to be included within the acknowledgement section of a paper. Guidance on how to acknowledge funding information is provided by the Research Information Network. The policy also requires all research papers, if applicable, to include a statement on how any underlying research materials, such as data, samples or models, can be accessed. However, the policy does not require that the data must be made open. If there are considered to be good or compelling reasons to protect access to the data, for example commercial confidentiality or legitimate sensitivities around data derived from potentially identifiable human participants, these should be included in the statement.

Acknowledgements: Acknowledgements should be the minimum consistent with courtesy. The wording of acknowledgements of scientific assistance or advice must have been seen and approved by the persons concerned. This section should not include details of funding.

-Please upload separate high quality figure files via the submission form.

-Author profile(s) must be uploaded via the submission form. Authors should submit a short biography (no more than 100

words for one author or 150 words in total for two authors) and a portrait photograph of the two leading authors on the paper. These should be uploaded, clearly labelled, with the manuscript submission. Any standard image format for the photograph is acceptable, but the resolution should be at least 300 dpi and preferably more. A group photograph of all authors is also acceptable, providing the biography for the whole group does not exceed 150 words.

-It is the authors' responsibility to obtain any necessary permissions to reproduce previously published material
https://jp.msubmit.net/cgi-bin/main.plex?form_type=display_requirements#use

END OF COMMENTS

Confidential Review

09-May-2022

Reviewing Editor:

The authors have done an excellent job of writing a topical review on scaling of neuronal spiking. Both reviewers agreed that the review was timely and addressed an important topic in the field. Both reviewers just had extremely minor issues that can be addressed with minor changes to the text. Please rewrite text taking into careful consideration of these points.

We were delighted to receive this positive evaluation of our submission. In the revised version we have addressed the points raised by the reviewers, as described below. The line number pointers refer to the revised manuscript.

Referee #1:

In this manuscript, the authors address the scaling of distributions of neuronal spiking within single neurons as well as across populations. Based on analyses of available experimental data and models they argue that a gamma distribution of firing rates is a better description than lognormal. They discuss "dark matter", irregular/ fluctuation-driven spiking, potential mechanisms of the distribution shape, and other interesting issues. The manuscript is somewhere in between a literature review and novel findings. Given the importance of the topic and the relatively sparse literature on the topic, I see the manuscript as an important contribution. I have a few comments.

Thank you for this highly favourable evaluation of our manuscript.

Section 2 is important but confusing:

"In our example, the log-transform makes both distributions bell-shaped (albeit left-skewed), revealing that the modes of the two log-distributions are similar but the orange distribution has heavier tails. "

First, there is no orange distribution in figure 2.

We agree that the colour is closer to brown than orange. To avoid such confusion, we edited the sentence (L74-75) and added to the figure a legend that labels each distribution.

What is the difference between 2B and 2C? Some more text in the caption and in the section would be helpful. I recommend writing more text details on the difference and how the measurements were analyzed to give the difference between these two panels. Is the data computer generated?

We have updated the text to explain the intuition behind the log-transform (L78-84). We now also include a statistical summary document which provides an example using histograms of datapoints drawn from the PDF, with the (MATLAB) code made publicly available on github.

Section 5: This section is important for explaining the mechanisms behind the skewed distributions. The argument is that a supra-linear I/O function turns normally distributed input (membrane potential) into firing rates that are not normally distributed (due to the nonlinearity). An argument suggested by Roxin et al, as the authors are well aware. The authors primarily justify this assertion by referring to previous modeling papers. But why not include the experimental work of Petersen and Berg 2016 in these references? Petersen and Berg investigated exactly this in the neuronal network dynamics by measuring membrane potential fluctuation and the supra-linearity across the population and the log-scaling of the spiking. They found that the input distribution is Gaussian and on average 3 sigmas from the threshold, both on single-cell level but also across the population.

Thank you for this excellent suggestion. We have updated the text to explicitly state that the theoretical work of Roxin et al. was verified experimentally by Petersen & Berg (L396-397).

6.1 "archi-cortical" -> "archicortical".

Fixed (L468), thank you.

Referee #2:

In the topical review "Logarithmically scaled, gamma distributed neuronal spiking", the authors discuss findings on the distributions of firing rates and inter-spike intervals (ISI) in, for the most part, cortical neurons. The main thrust of the review is that these distributions are far from Gaussian and are best considered on a logarithmic scale. Overall I believe this is a clear and readable introduction to the topic for the non-expert. This discussion of the differences between Gaussian and log-distributed (log-normal or log-gamma) distributions is well done, as is the discussion of the data. My only concerns would be regarding the details of mechanisms as understood through analysis of single-cell models and networks. I feel some of the details, which I think are of interest, are glossed over.

Thank you for this overall positive evaluation of the manuscript and the suggestion on the ways to improve it.

1 - Relationship between firing rate and CV: I feel Fig.3 could be explained more clearly. In particular, it would be nice to highlight the difference between the Poisson neuron and the LIF as far as the shape of the CV distribution for difference mean rates. As the authors state, the Poisson process always has $CV=1$, but this is not the case for the LIF and the ISI distribution narrows for increasing rates, a very general property of IF models and spiking networks in general. This is actually an open challenge in neuroscience since the CV for neurons in-vivo remains high even in the AS, e.g. Compte, Constantinidis et al. J. Neurophysiology 2003.

Thank you for raising this point. We revised the manuscript to explicitly point to the fact that the shape of log ISI distribution of IF neuron is not constant, unlike the Poisson neuron (L158-160).

We also considered adding two additional panels (reproduced below) to the figure. We decided that Fig. 3C provides a sufficient illustration but would be happy to get further feedback on this.

2 - Dale's principle and the fluctuation-driven regime: In section 5 the authors state that log-scaled ISIs and firing rates are somehow a consequence of Dale's Law. I have to say I don't understand this point. Dale's law state that each neuron is either E or I and cannot have an excitatory effect on some targets and an inhibitory effect on others. From the point of view of the post-synaptic neuron, the requirement for being in the fluctuation-drive regime is that E and I inputs should cancel more or less in the mean. This mechanism is agnostic as to where these inputs are coming from, and hence does not depend on Dale's law. At the network level, to find a self-consistent network state in a recurrent circuit in the fluctuation-driven regime may in some way depend on Dale's law, but it's not clear to me how. Once the network is in the fluctuation-driven regime then log-scaled rates and ISIs follow, yes.

We completely agree that the title of this section was misleading. We therefore changed it to “Log-scaled ISIs and mean firing rates from balanced input” and would like to thank the reviewer for pointing this out. In fact, it was not our intention to say that log-scaled ISIs and firing rates are the consequence of Dale’s principle. Rather we want to say that we can presume Dale’s principle and proceed based on this fundamental assumption, simply because these are the networks that one encounters.

What would have happened if Dale’s principle did not hold, i.e., if there was a single population of neurons that have both E & I synapses, or perhaps even individual synapses that are both E & I? Clearly in this case simple formalisms like Wilson-Cowan do not hold. It is possible that E-I balanced regimes would exist in such networks, but they probably would require some structural conditions on the number and strength of E & I synapses. This is an interesting theoretical question, which however is well outside the scope of the manuscript (because Dale’s principle does hold after all).

3 - Left-skewed log-rate distributions: The authors note that actual firing rate distributions are not exactly log-normal, but rather they have fatter left tails on a log-scale. They show that this is well fit by a log-gamma. It may also be of interest to show that the firing rate distributions calculated analytically from the theory for LIF neurons as well as the results from simulations of spiking neurons also reproduce this left-skewness and, in fact, can fit in-vivo data quite well, e.g. Roxin et al. J. Neurosci. 2011 Fig7.

Thank you for this suggestion. We fully agree that this is highly relevant to the points that the review is making, and have edited the text to explicitly make this point (L396-397).

Dear Michael,

Re: JP-TR-2022-282758R1 "Logarithmically scaled, gamma distributed neuronal spiking" by Daniel Levenstein and Michael Okun

I am pleased to tell you that your Topical Review article has been accepted for publication in The Journal of Physiology, subject to any modifications to the text that may be required by the Journal Office to conform to House rules.

NEW POLICY: In order to improve the transparency of its peer review process The Journal of Physiology publishes online as supporting information the peer review history of all articles accepted for publication. Readers will have access to decision letters, including all Editors' comments and referee reports, for each version of the manuscript and any author responses to peer review comments. Referees can decide whether or not they wish to be named on the peer review history document.

The last Word version of the paper submitted will be used by the Production Editors to prepare your proof. When this is ready you will receive an email containing a link to Wiley's Online Proofing System. The proof should be checked and corrected as quickly as possible.

All queries at proof stage should be sent to tjp@wiley.com

The accepted version of the manuscript will be published online, prior to copy editing in the Accepted Articles section.

Are you on Twitter? Once your paper is online, why not share your achievement with your followers. Please tag The Journal (@jphysiol) in any tweets and we will share your accepted paper with our 22,000+ followers!

Best wishes

Katalin Toth
Senior Editor
The Journal of Physiology

* IMPORTANT NOTICE ABOUT OPEN ACCESS *

To assist authors whose funding agencies mandate public access to published research findings sooner than 12 months after publication The Journal of Physiology allows authors to pay an open access (OA) fee to have their papers made freely available immediately on publication.

You will receive an email from Wiley with details on how to register or log-in to Wiley Authors Services where you will be able to place an OnlineOpen order.

You can check if your funder or institution has a Wiley Open Access Account here <https://authorservices.wiley.com/author-resources/Journal-Authors/licensing-and-open-access/open-access/author-compliance-tool.html>

Your article will be made Open Access upon publication, or as soon as payment is received.

If you wish to put your paper on an OA website such as PMC or UKPMC or your institutional repository within 12 months of publication you must pay the open access fee, which covers the cost of publication.

OnlineOpen articles are deposited in PubMed Central (PMC) and PMC mirror sites. Authors of OnlineOpen articles are permitted to post the final, published PDF of their article on a website, institutional repository, or other free public server, immediately on publication.

Note to NIH-funded authors: The Journal of Physiology is published on PMC 12 months after publication, NIH-funded authors DO NOT NEED to pay to publish and DO NOT NEED to post their accepted papers on PMC.

EDITOR COMMENTS

Reviewing Editor:

The authors have done an excellent job of responding to the reviewers and there are no further concerns.